# On the warm bias in atmospheric reanalyses induced by the missing snow over Arctic sea-ice

Yurii Batrak [1] & Malte Müller [1]

Over the past decades, the Arctic has been warming more than any other region in the world with profound socio-economic consequences. One of the key elements for understanding this rapid climate change is the surface energy budget. However, in the Arctic this budget is not consistently described across the various climate models, reanalyses and observation products. Recognising the physical causes of these inconsistencies is highly relevant for improving climate predictions and projections. Here we show that a 5 to 10 °C warm bias of the sea-ice surface temperature in global atmospheric reanalyses and weather forecasts is mainly caused by a missing representation of the snow layer on top of the sea-ice. Due to the low thermal conductivity of snow compared to sea-ice, a thin snow layer reduces the conductive heat flux much more efficiently than sea-ice, and thus insulates the cold atmosphere from the relatively warm ocean.

---

[1] Development Centre for Weather Forecasting, Norwegian Meteorological Institute, Henrik Mohns Plass 1, 0313 Oslo, Norway. Correspondence and requests for materials should be addressed to M.M. (email: maltem@met.no)

There is a growing demand for a more accurate prediction and understanding of Arctic weather and climate, which puts great expectations on the global and regional forecasting and reanalysis systems[1]. Currently, weather and climate models are subjected to having various systematic errors, which induce large spread in results of different climate models[2] and have a significant impact on the local and mid-latitude prediction skill[3]. Also, contemporary global reanalysis products, often used in climate research in order to monitor climate change and variability, or to evaluate climate model systems[4], have large biases in temperature, humidity and wind speed in the Arctic[5–7].

Specifically, in the Arctic surface energy balance significant deviations between climate models are found[2,4]. Within the coupled atmosphere, ocean, snow, and sea-ice system, this energy balance controls the growth and melt of sea-ice, as well, as the thermal stratification of the lower atmosphere. Thus, the accurate simulation of the Arctic surface energy budget is key for improving our understanding of the rapid climate change in the Arctic, as well as, advancing long term prediction of sea-ice properties in the future.

During Arctic winter clear-sky events (CSE) over sea-ice, with the absence of solar radiation and the strong longwave radiative cooling, the air temperatures can drop to −40 °C. Studies of the winter Arctic surface energy budget show that the radiative cooling of the troposphere is balanced by the advection of heat from lower latitudes[8,9]. In addition, in sea-ice covered regions the ocean moderates the low temperatures during CSEs in contrast to the land surfaces, where minimum temperatures of −50 to −60 °C occur[8].

In the present study, we investigate the ability of contemporary regional and global weather prediction systems and global reanalysis systems to simulate the low temperatures observed during CSEs. We use the in-situ data set from the N-ICE 2015 drift campaign[10] and a pan-Arctic sea-ice surface temperature satellite product. We find a warm bias in almost all analysed model systems and show that it is induced by a missing representation of snow on sea-ice. Furthermore, the simplistic representation of sea-ice thickness and concentration in the analysed model systems contribute to inconsistencies in the simulation of sea-ice surface temperatures.

## Results

**Overview of the studied models and utilised observations.** We use the observational in situ data set of atmospheric, snow, sea-ice and ocean observations which is available from the N-ICE 2015 campaign and taken during the four subsequent drifts of a research vessel in between January and June 2015 (Fig. 1). The meteorological conditions, the thermodynamic structure of the troposphere, and the surface energy budget during the campaign are analysed in a set of studies[11–13]. Several CSE have been observed during N-ICE campaign[13]. In addition, we utilise a pan-Arctic sea-ice surface temperature satellite product[14] based on infrared data from AVHRR instruments in order to extend our results to the pan-Arctic scale. More details on the use of the observations are given in the Methods section.

We investigate the ability of regional and global weather prediction systems (HARMONIE-AROME[15] configuration of the ALADIN-HIRLAM numerical weather prediction system, and IFS-HRES[16]), and global reanalysis systems (ERA-Interim[17], ERA5, MERRA-2[18], JRA-55[19], and NCEP-DOE Reanalysis 2[20]) to simulate those low temperature events. For HARMONIE-AROME two configurations of the sea-ice parameterisation scheme are studied. First, the control experiment (AA) which resembles sea-ice handling in IFS-HRES and has a snow-free sea-ice layer of uniform and fixed thickness. Second, the sensitivity

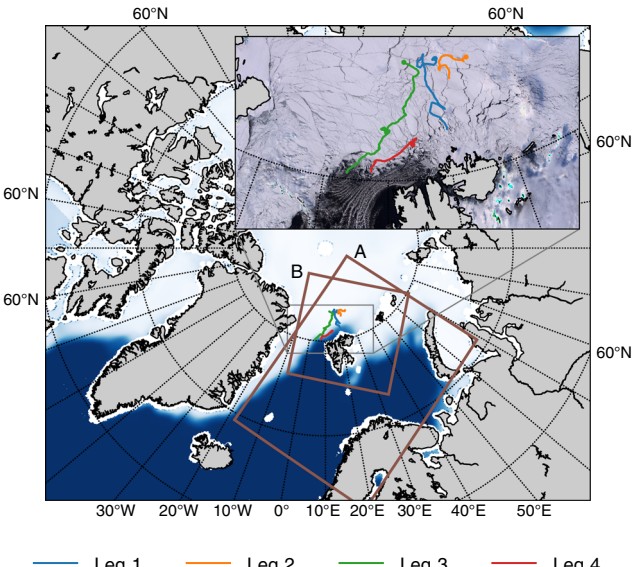

**Fig. 1** General overview of the sea-ice cover, experimental model domains and N-ICE 2015 campaign drifts. Background: mean sea ice concentration in January–February 2015 according to OSI-SAF data; A: model domain for the AA (control) experiment, B: model domain for the AA-snow (sensitivity) experiment; coloured curves: trajectories of the different N-ICE 2015 campaign legs; inset: trajectories of N-ICE 2015 drifts and restored true colour image retrieved from MODIS radiances showing the ice cover on 3 April 2015. Leg 1: 15 January–21 February; leg 2: 24 February–19 March; leg 3: 18 April–5 June; leg 4: 7 June–22 June

experiment (AA-snow) which uses extended sea-ice parameterisation scheme that includes a prognostic sea-ice thickness and snow layer model (see the Methods section for further details). General overview of the model systems discussed in the present study is provided in the Table 1.

**Analysis of a clear-sky event from N-ICE 2015.** During the winter drifts of the N-ICE 2015 campaign a large number of storms has been observed[11]. Since those storms are connected to the advection of warm air onto the sea-ice, they can be identified in the temperature timeseries by relatively high temperatures close to 0 °C (Fig. 2). In between those storms, periods of clear-sky conditions occurred, marked by reduced longwave downwelling radiation flux (LW↓) of around 160 Wm$^{-2}$ and low temperatures of around −30 to −40 °C. We refer in the following to three clear-sky events, CSE-1, CSE-2, and CSE-3, which are marked in Fig. 2.

During the CSE-1 all models showed low LW↓ values of 140 to 150 Wm$^{-2}$, which are consistent with the observations (see the Fig. 2) and indicate that clear-sky conditions are well simulated. However, the simulated surface temperatures ($T_{skin}$) are much higher than observed. While the observations show values of down to −40° C, most of the models simulate temperatures of 5 to 15° C too warm, with the strongest deviations found in the snow-free AA experiment. The sensitivity experiment AA-snow, with a prognostic sea-ice thickness and snow layer model, is the only model simulation with small deviations. For all model products, the $T_{skin}$ is obtained from the LW↓ and LW↑, while LW↑ corresponds to the sea-ice covered part of the grid cell only (see also Eq. (6) in the Methods section).

During the CSE-2 and CSE-3, the models show a similar warm temperature bias as during the CSE-1, but with differences in their capabilities in simulating the low LW↓, which are characteristic for clear-sky conditions. During CSE-2, MERRA-2

**Table 1 Overview of the numerical weather prediction systems and global reanalysis systems**

|  | AA | AA-snow | IFS-HRES | MERRA-2 | ERA5 | ERA-I (Interim) | JRA-55 | NCEP-2 |
|---|---|---|---|---|---|---|---|---|
| Weather model | HARMONIE-AROME cy40h1.1 |  | IFS cy40r1 | GEOS 5.12.4 | IFS cy41r2 | IFS cy31r2 | JMA-GSM0603 | modified GFS MRF95 |
| Horizontal resolution |  |  | T1279 | 0.5° × 0.6° | T639 | T255 | T319 | T62 |
|  | 2.5 km | 2.5 km | 16 km | 65 km | 31 km | 79 km | 55 km | 210 km |
| Vertical levels | 65 | 65 | 137 | 72 | 137 | 60 | 60 | 28 |
| Sea ice thickness | 0.75 m | Prognostic | 1.5 m | n/a[a] | 1.5 m | 1.5 m | 2 m | 2 m |
| Sea ice cover | Fractional | Fractional | Fractional | Fractional | Fractional | Fractional | Binary | Binary |
| Snow on sea ice | No | Yes | No | No | No | No | No | Yes[b] |

[a]Ice temperature is resolved by means of thermal balance of a single ice layer of 7-cm thickness. Obtained temperature is relaxed towards 0 °C to take account of the upwelling ocean heat flux
[b]Snow has no effect on the thermal conductivity of the sea-ice layer

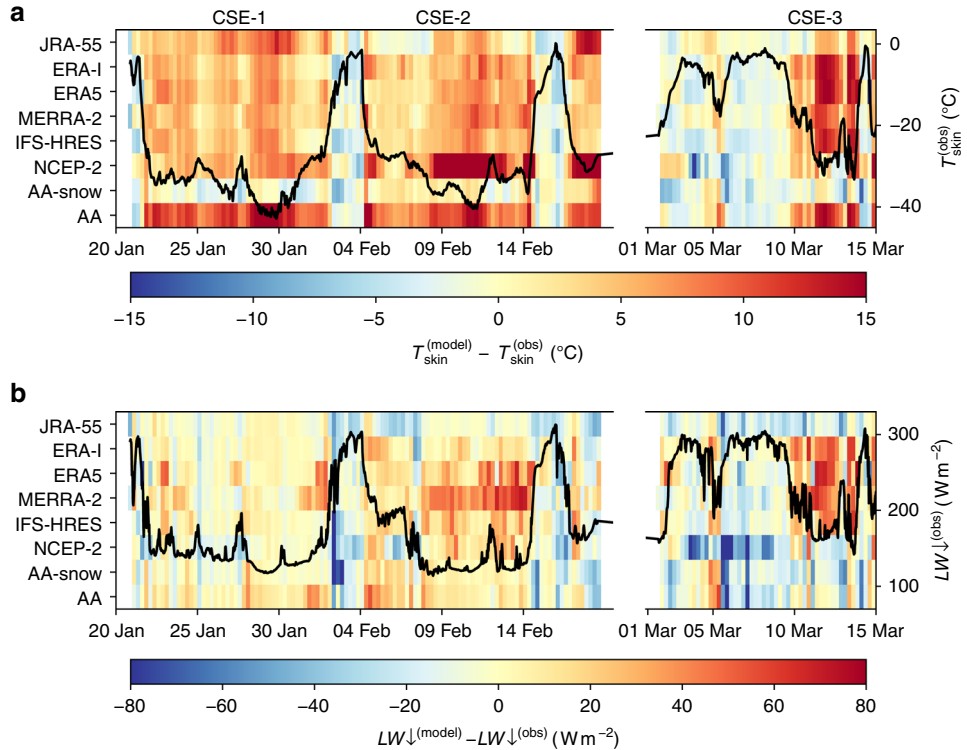

**Fig. 2** Observations from the N-ICE 2015 campaign compared to various reanalysis products and model experiments. **a** Difference between the skin temperature predicted by different models and skin temperature observed during the N-ICE 2015 campaign (background); time series of the measured skin temperature (foreground curve). **b** Same as **a** but for downwelling longwave radiation

and ERA5 show too high LW↓ indicating that the clear sky conditions are not well resolved. During CSE-3, the ECMWF products (ERA5, ERA-I, and IFS-HRES) and MERRA-2 do not show the resolved clear-sky conditions and provide too high LW↓. For this event, the temperature biases are largest in the ECMWF products with the values of up to 15 °C.

In general, the models show a consistent warm bias during the polar winter clear-sky conditions over sea-ice, when observed temperatures are usually below −25 °C. In order to better understand this discrepancy between the modelled and observed temperatures, we analyse the surface energy budget (see also the Methods section) during the CSE-1, where all models show a consistent representation of the clear sky conditions in terms of the simulated low LW↓ values. Averaged values of the surface energy budget are given in Table 2. The conductive heat flux $C$ is

calculated (assuming steady linear sea-ice temperature profile) by using the respective snow and sea-ice thickness properties of the models (Table 1) or, in case of AA-snow, the directly simulated daily values of the sea-ice and snow thickness. Note that, for the sake of convenience, we assumed a 1.5-m sea-ice thickness for the MERRA-2 reanalysis. We tested the sensitivity of the flux by also prescribing sea-ice thicknesses of 1 and 2 m, which resulted in conductive heat fluxes ranging from 30 to 61 Wm$^{-2}$, i.e., in the range of the values obtained from the other model systems. In order to compute the observed conductive heat flux the mean observed values for sea ice and snow thickness are used[21], i.e., $h_s = 0.5$ m and $h_i = 1.4$ m.

The conductive heat flux in the observations is 16 Wm$^{-2}$. The AA-snow experiment shows a conductive heat flux consistent with the observations. The AA experiment shows a much higher

**Table 2 Surface energy budget–Clear sky event 1**

|  | OBS | AA | AA-snow | IFS-HRES | MERRA-2 | ERA5 | ERA-I | JRA-55 | NCEP-2 |
|---|---|---|---|---|---|---|---|---|---|
| $C$, $Wm^{-2}$ | 16 | 61 | 15 | 38 | 40 | 37 | 39 | 28 | 32 |
| LW↑, $Wm^{-2}$ | −182 | −221 | −179 | −199 | −197 | −201 | −198 | −203 | −199 |
| LW↓, $Wm^{-2}$ | 142 | 148 | 140 | 146 | 144 | 145 | 140 | 138 | 132 |
| LW↑ + LW↓, $Wm^{-2}$ | −40 | −73 | −39 | −53 | −53 | −56 | −58 | −65 | −67 |

flux of 61 $Wm^{-2}$, which is due to the combination of a missing snow layer and a small prescribed sea-ice thickness of 0.75 m. The global reanalyses, which do not represent snow over sea-ice (NCEP-2 has a rudimentary parameterisation for snow over sea-ice, but it could be neglected. See Table 1 and Methods for further details) but have a realistic assumption of the sea-ice thickness of 1.5 to 2 m, still overestimate the conductive heat flux by a factor of more than two, with values between 28 and 40 $Wm^{-2}$. The JRA-55 and NCEP-2 have the smallest values, due to their relatively large sea-ice thickness of 2 m.

The overestimation in $C$ from the ocean to the surface is to a large extent compensated by an increased LW↑. The observed value for LW↑ is 182 $Wm^{-2}$, while the global reanalyses show values of around 200 $Wm^{-2}$ and in the AA experiment the flux reaches 220 $Wm^{-2}$. Again, AA-snow is consistent to the observations with a value of 179 $Wm^{-2}$.

We conclude that the missing snow layer on top of the sea-ice results in the overestimated conductive heat flux from the ocean to the atmosphere, which is compensated by an increase in outgoing longwave radiation and too high surface temperatures. The resulting bias in the net surface radiation budget (LW↑ + LW↓) is about 20 to 30 $Wm^{-2}$.

**A pan-Arctic view of the temperature bias**. From infrared satellite observations we can estimate the pan-Arctic $T_{skin}$ during winter clear-sky conditions in the selected period from 2015 to 2017. In the satellite observations the mean clear-sky surface temperatures are as low as −35 °C in large parts of the central Arctic and towards the Canadian coastline (Fig. 3a).

In the coldest areas, all reanalyses have a warm bias of about 5 to 10 °C (Fig. 3e–f). ERA5 (Fig. 3c) and MERRA-2 (Fig. 3e) have similar spatial characteristics for the surface temperature with both showing a warm bias in all sea-ice covered areas and MERRA-2 having the largest temperature biases. The JRA-55 (Fig. 3d) and NCEP-2 (Fig. 3f) reanalyses have a smaller warm bias in the central Arctic and tend to have a cold bias in the areas further away from the North pole. This cold bias is about 5 to 10 °C and is most pronounced in the Northern Barents Sea, the areas from Hudson Bay towards Baffin Bay and some parts along the Russian, Canadian and American coastlines. Note, here we derive the model simulated surface skin temperature from the LW↓ and LW↑, following the Eq. (1) in the Methods section. This is equivalent to the grid cell mean temperature of the model, and not the surface temperature of the sea-ice part of the grid cell only, and thus consistent to the infrared surface observation.

The pattern of the $T_{skin}$ warm bias in the global reanalyses shows similar characteristics to the retrieved mean snow depth from the TOPAZ4 ocean and sea-ice reanalysis[22] during CSEs in 2015 to 2017 (Fig. 3b). For example the lower snow-depths in Baffin Bay and Labrador Sea result in higher observed $T_{skin}$ and a smaller bias, while the largest snow-depths in the central Arctic and towards the Canadian Archipelago are co-existing with the lowest temperatures and the largest temperature biases in the models.

Our findings of a temperature bias during clear sky conditions in the satellite analysis are consistent to results of the N-ICE 2015

drift analysis in the previous section. In order to further strengthen the conclusion that the snow representation is the key factor, the surface temperature bias can be formulated as a function of the misrepresentation of snow and the sea-ice thickness (see Methods section and Fig. 4). If the prescribed sea-ice thickness is 1.5 m, an error in the snow thickness of only 0.25 m can induce a warm bias of 8 °C. This could be seen in ERA5 and MERRA-2 over the central Arctic (see Fig. 3c and e) where largest warm biases of the surface temperature are in a good agreement with the pattern of the mean snow depth (Fig. 3b). A misrepresentation in the sea-ice thickness can also have a strong effect on the surface temperature in cases of only a thin snow layer. The JRA-55 and NCEP-2 reanalyses have a 2 m sea-ice thickness representation, compared to 1.5 m in ERA5 and MERRA-2. If we assume that there is a 0.5 m overestimation in sea-ice thickness in regions with no snow cover, it would lead to a cold bias of about −5 °C, and counter-balance the warm bias induced by the missing snow in areas with a snow layer of around 0.1 m (Fig. 4). This is consistent with the relatively small warm bias in central parts of the Arctic in JRA-55 and NCEP-2, and agrees with the cold bias further away from the North pole in these reanalyses (see the Fig. 3d and f). In addition, JRA-55 and NCEP-2 have a binary representation of the sea-ice cover. When sea-ice concentration is above 55% (or 50% for NCEP-2) these reanalyses assume that sea is completely ice-covered. This induces the additional spurious insulation from the ocean and, in turn, a cold surface temperature bias in regions closer to the sea-ice edge.

**Discussion**
An evaluation of the surface energy budget of current global and regional model configurations, which are used for weather fore-casting and reanalysis products, highlights the importance of the simulation of snow on sea-ice. Three CSEs occurred during polar night are analysed using the N-ICE 2015 drift campaign data. The analysis shows that the models in their standard configurations are 5 to 15 °C too warm during these events. In the CSE-1 the low incoming longwave radiation in the model products is low, which indicates that the clear-sky conditions are reasonably well resolved. However, the outgoing longwave radiation is too high, as the consequence of a warm bias in the surface temperature. In turn, this bias can be attributed to the overestimation of the conductive heat flux from the ocean to the surface by 20 to 40 $Wm^{-2}$. This heat flux is regulated by the difference between the ocean and atmospheric temperatures and by the thickness of the sea-ice and snow layers. A snow layer, which has a thermal conductivity about seven times lower than that of sea-ice, is missing in most of the analysed models, and thus, is identified as the main reason for the deviations in the surface energy budget.

The warm bias in the ice surface temperatures is a character-istic feature of the analysed global reanalysis products on the pan-Arctic scale. By comparing to a multiyear product obtained from infrared satellite measurements we show that all tested systems have issues in simulating the extreme low temperatures over large parts of the Arctic. The most drastic deviations in the tempera-ture are found over the areas with the thick snow cover according to the winter snow climatology (see the Fig. 3b–f). This

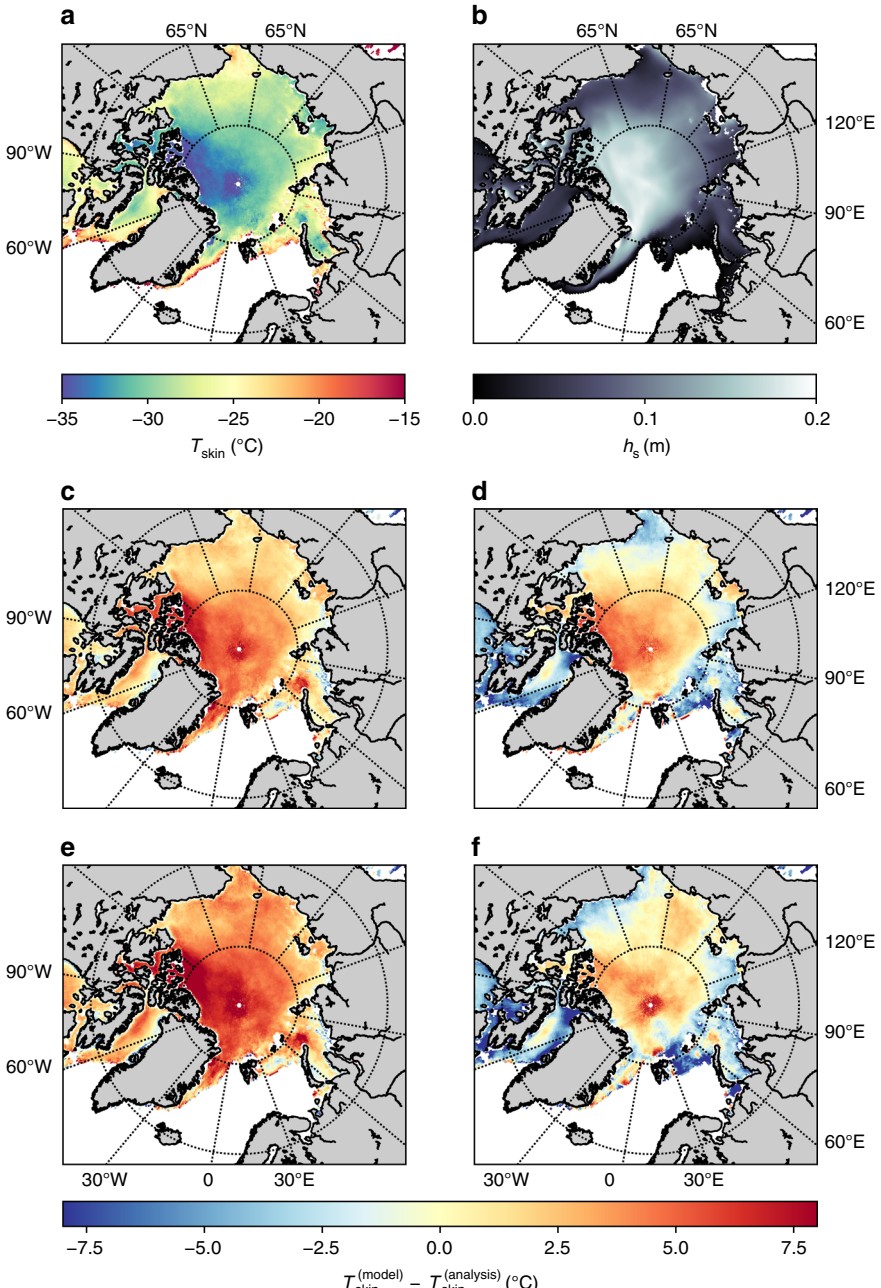

**Fig. 3** Satellite ice surface temperature product compared to various reanalysis products. **a** Climatological winter-time clear-sky ice surface temperature in CMEMS for years 2015–2017; **b** Climatological snow thickness according to the TOPAZ reanalysis for years 2015–2017; **c**–**f** difference between the climatological winter-time clear-sky ice surface temperature in different reanalysis products and the climatological winter-time clear-sky ice surface temperature in CMEMS for years 2015–2017, (**c**) ERA5, (**d**) JRA-55, (**e**) MERRA-2, (**f**) NCEP-2

strengthens the argument that the snow layer is strongly connected to the temperature bias found in the global atmospheric reanalyses. In two of the reanalyses, which have thicker sea-ice (JRA-55 and NCEP-2), the warm bias is reduced, but instead a cold bias is produced, specifically in areas where sea-ice is thinner than the prescribed 2 m and only a thin snow layer exists. In general, the characteristics of the surface temperature in the reanalysis products can be explained by the surface energy budget. We conclude that the snow component on the sea-ice improves the surface atmospheric energy budget in cold atmospheric conditions and thus is an important but often missing component in state-of-the-art reanalysis and forecasting systems.

In addition, sea-ice thickness plays an important role, as well, but mainly in the areas where the snow layer is thin.

The surface energy budget is an integral part of many climate processes in the Arctic. For example, it determines the available energy for sea-ice melting and freezing, and also the thermal stratification in the lower troposphere. Thus, the accuracy of its representation can have a strong impact on the skill of climate prediction and our understanding of large-scale climate dynamics. The bias in the surface energy budget due to the misrepresentation of the snow and sea-ice layer is about 20 Wm$^{-2}$. Compared to the spread of 20 and 60 Wm$^{-2}$ between the various climate models in the winter-time net longwave radiation

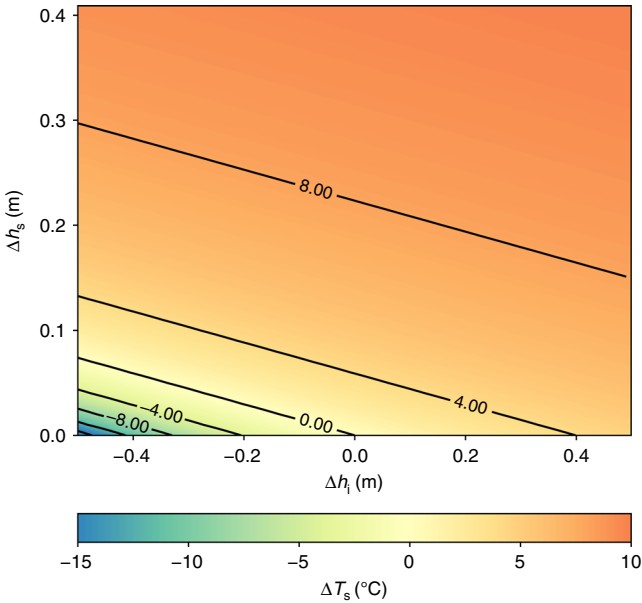

**Fig. 4** The dependency of the surface temperature bias on ice and snow thickness errors. The bias $\Delta T_s$ is induced by the errors of ice $\Delta h_i$ and snow thickness $\Delta h_s$. Here, we assume a simulated 1.5 m thick sea-ice without a snow layer, i.e., $h_i = 1.5$ m and $h_s = 0$ m

budget[2,4], the temperature bias, which is induced by the missing snow layer and misrepresentation of sea-ice thickness, is of significant importance. This bias in the net radiation budget can induce wrong conclusions, e.g., assuming that the climate models have a "cold temperature" bias in winter[4], or generally refine process studies which rely on the atmospheric reanalyses [9,23].

Due to the widespread use of the atmospheric reanalysis products for model validation, initialisation of prediction systems, forcing of ocean and sea-ice reanalyses[24], etc., it is very important to take into account this temperature bias in the contemporary reanalyses. Future reanalyses could be enhanced primarily by considering a prognostic parameterisation scheme to represent snow cover over the sea-ice, and by improving the spatial characteristics of the sea-ice thickness.

## Methods

**Observations**. We use the observational data set from the N-ICE 2015 drift north of the Svalbard Archipelago[10]. The N-ICE 2015 campaign consisted in total of four legs. In the present study we focus on the first two legs within the Arctic winter period, between 15 January and 19 March (Fig. 1), and analyse the meteorological observations of longwave radiation[11]. The skin temperature of the snow surface is obtained from the longwave upwelling LW↑ and downwelling LW↓ radiation by

$$T_{\text{skin}} = \left(\frac{\text{LW}\uparrow - (1 - \varepsilon_s)\text{LW}\downarrow}{\varepsilon_s \sigma}\right)^{\frac{1}{4}}, \tag{1}$$

where $\sigma$ is the Stefan–Boltzmann constant and $\varepsilon_s$ is the emissivity of the snow, which we assume to be 0.99[13].

In addition to the in-situ observation of N-ICE 2015, we include a satellite data set of daily sea-ice surface temperature ($T_{\text{skin}}$). The data product is based on infrared data from the AVHRR instrument on board the Metop-A satellite. The $T_{\text{skin}}$ satellite data set is available from 2015 to present, as part of the Copernicus Marine Environmental Monitoring Service (CMEMS). Since satellite measurements of $T_{\text{skin}}$ are only meaningful in case of cloud free conditions, for the satellite retrieval a cloud-mask has been used together with a bias-correction algorithm to obtain surface temperature estimates within cloud covered areas[14]. In the present study, however, we excluded those corrected surface temperatures, due to our focus on CSEs in winter. We define the CSEs by considering only the data points where: incoming longwave radiation in ERA5 is lower than 160 Wm$^{-2}$, the OSI SAF Ice Edge product identifies the grid cell as closed ice, there are more than eight AVHRR infrared observations in the grid cell, and the date is within the winter season (October to February). The same mask is then applied to the daily surface temperature values, in order to derive the mean $T_{\text{skin}}$ of reanalysis and observations during clear sky conditions on a pan-Arctic scale.

**Atmospheric models and global reanalysis products**. We analyse a suite of global reanalyses, i.e., two configurations of the ECMWF model system (ERA-Interim[25] and ERA5), the Modern-Era Retrospective analysis for Research and Applications, Version 2 (MERRA-2[18]), the Japanese 55-year Reanalysis (JRA-55[19]), and the National Centers for Environmental Prediction Reanalysis 2 (NCEP-2[20]).

We also include the operational ECMWF global deterministic weather forecasting system (IFS-HRES[16]), as well as, two experimental configurations of the regional convective scale model HARMONIE-AROME[15]. The first of these configurations (AA) covers the same domain as the operational numerical weather prediction system AROME Arctic[26] and has the same representation of the sea-ice cover. The second experimental configuration of HARMONIE-AROME (AA-snow) is similar to AA, but incorporates the parameterisations for the sea-ice mass-balance and snow layer on top of the sea-ice. The model domain for AA-snow is smaller than that one in the AA experiment (see the Fig. 1) to reduce the computational costs and to focus on the areas with extensive sea-ice cover. The AA experiment covers the time period from 10 January to 20 March 2015. The AA-snow experiment has been performed from 1 September 2014 to 20 March 2015. To allow realistic evolution of the snow cover in AA-snow, this experiment was started without snow layer on top of sea ice and snow was accumulated from the modelled precipitation. In both AA and AA-snow the initial ice thickness is 0.75 m but in AA it remains constant during the experiment and in AA-snow it evolves according to the ice mass-balance parameterisation.

Relevant features of the models, which were used to produce forecast and reanalysis products, are given in Table 1 and with a focus on the representation of the sea-ice surface in the Arctic. In short, none of the global model systems has a prognostic parameterisation of the sea-ice mass-balance nor a parameterisation of a snow layer on top of the sea-ice, which impacts the surface energy budget. NCEP-2 simulates snow on sea-ice, however, without any influence on the heat conductivity and surface albedo. In all models sea-ice concentration is updated daily by utilising satellite products. Operational configurations of HARMONIE-AROME do not simulate the sea-ice concentration and sea-ice thickness, but simulate sea-ice surface temperature[26,27].

To extend the existing sea-ice parameterisation of HARMONIE-AROME with ice mass-balance calculations, we added the representation of the following processes: ice growth and melting from the bottom, and surface melting. The more complex processes of internal melting and snow-ice formation were not considered.

The interface between the sea-ice bottom and the underlying water body is governed by the balance between the sea-ice and ocean heat fluxes. When these fluxes are not in equilibrium state, the residual heat flux leads to growth or melting of the ice layer[28]:

$$\rho_i L_f \frac{dh_i}{dt} = k_i \frac{\partial T_i}{\partial z}\bigg|_{z=h_i} - F_w \tag{2}$$

where $\rho_i$ is the density of sea-ice, $L_f$ is the latent heat of fusion, $h_i$ is the total ice thickness, $k_i$ is the thermal conductivity of sea-ice, and $F_w$ is the ocean heat flux. For the AA-snow experiment, the ocean heat flux was assumed to be constant with the value of 2 Wm$^{-2}$.

The processes of surface melting is parameterised in the following way. At the first step, the ice surface temperature is calculated from the thermal balance of the surface layer of sea-ice. Then, if the obtained temperature is higher than the melting temperature of sea-ice, it is set to be equal this melting temperature and the residual heat flux $F_{\text{melt}}$ induces the melting of the ice surface[28]:

$$F_{\text{melt}} = -\rho_i L_f \frac{dh_i}{dt} \tag{3}$$

The AA-snow experiment, which uses the updated sea-ice parameterisation scheme, shows simulated snow depths between 0.45 and 0.7 m and sea-ice thickness between 1.4 and 1.7 m. This is consistent with the observed snow and sea-ice thickness of $0.5 \pm 0.2$ m and $1.4 \pm 0.3$ m, respectively[21].

We compare the model's $T_{\text{skin}}$ with the satellite and in-situ observations described in the previous sections. For the satellite observations $T_{\text{skin}}$ corresponds to the model's grid-cell mean temperature, and in case of the in-situ observations it corresponds to the $T_{\text{skin}}$ of the sea-ice covered part only. Thus, for the satellite observation comparison we can use the Eq. (1). For the comparison with the in-situ observations we partition the longwave upwelling radiation LW↑ into an ocean and sea-ice covered part according to the sea-ice concentration SIC at the respective grid-cell.

$$\text{LW}\uparrow = (1 - \text{SIC}) \cdot \text{LW}\uparrow^{(\text{ocean})} + \text{SIC} \cdot \text{LW}\uparrow^{(\text{sea-ice})} \tag{4}$$

If we assume the ocean surface with temperature of $-2°$ C and emissivity of 0.98, then by following the Stefan-Boltzman law we obtain

$$\text{LW}\uparrow^{(\text{ocean})} = 0.98 \cdot \sigma \cdot 271.15^4 \cdot \tag{5}$$

Thus, combining the Eqs. (1), (4), and (5), the skin temperature of the ice covered

part of the grid-cell is derived as follows:

$$T_{\text{skin}}^{(\text{sea-ice})} = \left[ \frac{\text{SIC}^{-1}\left(\text{LW}\uparrow - (1-\text{SIC})\text{LW}\uparrow^{(\text{ocean})}\right) - (1-\varepsilon_s)\text{LW}\downarrow}{\varepsilon_s \sigma} \right]^{\frac{1}{4}} \quad (6)$$

**Surface energy budget and surface temperature.** The energy budget of the surface layer, or the net-energy transfer $F_{\text{sfc}}$ between the atmosphere and the ocean can be written as[9,13]

$$F_{\text{sfc}} = C_t \frac{\partial T_s}{\partial t} - C = (1-i_0)(1-\alpha)\text{SW}\downarrow + \text{LW}\downarrow + \text{LW}\uparrow + Q \quad (7)$$

where $C_t$ is the thermal resistance of the surface layer, $C$ is the the conduction of heat from the ocean through the sea-ice/snow to the atmosphere, $i_0$ denotes the part of the downwelling shortwave radiation that penetrated through the surface layer, $\alpha$ is the surface albedo, $\text{SW}\downarrow$ is the surface downwelling short-wave radiation flux, $\text{LW}\downarrow$ is the surface downwelling longwave radiation flux, $\text{LW}\uparrow$ is the surface upwelling long-wave radiation flux, $Q$ is the turbulent flux of sensible and latent heat. In the polar night conditions the downwelling shortwave radiation flux is very small and could be neglected from the Eq. (7). The conductive flux from the ocean to the snow surface is not directly observed. However, we can estimate it as

$$C = \frac{k_i \cdot k_s}{k_s \cdot h_i + k_i \cdot h_s}(T_o - T_{\text{skin}}) \quad (8)$$

where $T_o$ is the surface ocean temperature, $k_s$ is the thermal conductivity of snow, and $h_s$ is the thickness of the snow layer on top of sea-ice. For the thermal conductivity of snow and sea-ice we assume, $k_s = 0.31\ \text{Wm}^{-1}\,\text{K}^{-1}$ and $k_i = 2.1\ \text{Wm}^{-1}\,\text{K}^{-1}$, respectively. Note that Eq. (8) implies steady linear temperature profile within the sea-ice layer. This assumption is not generally correct in case of rapid changes of weather conditions or multilayer sea-ice schemes, especially when snow and ice layers are of considerable thickness. However, taking into account minor variability of weather conditions during a single polar night CSE, Eq. (8) could provide a simple first-order estimate of the real conductive heat flux.

If we assume that a variation $\Delta R$ in the net longwave radiation budget $R = \text{LW}\downarrow + \text{LW}\uparrow$ balances the changes in the conductive heat flux $\Delta C$ induced by errors in snow thickness $\Delta h_s$ and sea ice thickness $\Delta h_i$, we can write $\Delta R = \Delta C$. For the radiation we yield

$$\Delta R = R(T_s + \Delta T_s) - R(T_s) = \varepsilon\sigma \cdot (T_s + \Delta T_s)^4 - \varepsilon\sigma T_s^4 \approx 4\varepsilon\sigma T_s^3 \cdot \Delta T_s \quad (9)$$

with $\Delta T_s$ being the induced temperature bias. And the variation of the conductive heat flux is defined as

$$\Delta C = C(T_s + \Delta T_s, h_i + \Delta h_i, h_s + \Delta h_s) - C(T_s, h_i, h_s) \quad (10)$$

$$= \alpha' \cdot (T_o + \Delta T_s - T_{\text{skin}}) - \alpha \cdot (T_o - T_{\text{skin}}) \quad (11)$$

where

$$\alpha \equiv \frac{k_i \cdot k_s}{k_s \cdot h_i + k_i \cdot h_s} \quad (12)$$

$$\alpha' \equiv \frac{k_i \cdot k_s}{k_s \cdot (h_i + \Delta h_i) + k_i \cdot (h_s + \Delta h_s)} \quad (13)$$

Thus, we derive the temperature bias induced by changes in snow and sea-ice thickness during clear sky conditions as

$$\Delta T_s = \frac{(\alpha' - \alpha) \cdot (T_s - T_o)}{4\varepsilon\sigma T_s^3 - \alpha'}. \quad (14)$$

## Data availability

The data from the N-ICE 2015 drift is available at the Norwegian Polar Data Centre (https://doi.org/10.21334/npolar.2016.7f7e56d0). The sea-ice surface temperature satellite data product set is available from the Copernicus Marine Environmental Monitoring Service (http://marine.copernicus.eu). The reanalysis products are accessible via the respective dissemination units. Data generated and analysed during the current study are available from the corresponding author on reasonable request.

## Code availability

ALADIN-HIRLAM numerical weather prediction system is developed in cooperation between the ALADIN and HIRLAM consortia and not available to general public. A copy of the source code of the ALADIN-HIRLAM numerical weather prediction system could be obtained for non-commercial research purposes from a member institution of ALADIN or HIRLAM consortium in applicant's country after signing a standardised license agreement (http://www.hirlam.org/index.php/hirlam-programme-53/access-to-the-models).

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

## Acknowledgements
The authors would like to thank Wesley Ebisuzaki, Shinya Kobayashi, Ron Gelaro, and
Richard Cullather for their valuable comments concerning the details of representation of the
sea-ice cover in the reanalysis systems. We acknowledge the open data policy of the N-ICE
project. This research was funded by the Research Council of Norway through the Nansen
Legacy project (NFR-276730). This is a contribution to the Year of Polar Prediction (YOPP),
a flagship activity of the Polar Prediction Project (PPP), initiated by the World Weather
Research Programme (WWRP) of the World Meteorological Organization (WMO).

## Author contributions
The main idea was developed by Y.B. and M.M. Y.B. developed, designed, and performed
the snow on sea-ice model experiments, and produced the figures. M.M. performed the
analysis of the observations, model experiments, and reanalysis products. Both authors
contributed to the discussions, interpretation of the results and writing the paper.

## Additional information

**Competing interests:** The authors declare no competing interests.

