## [Peer Review File · Nature Communications]

Reviewers' comments:

Reviewer #1 (Remarks to the Author):

Review of the manuscript "On the warm bias of atmospheric reanalyses over Arctic sea-ice in winter" by Yurii Batrak and Malte Muller.

The authors report that several widely-used atmospheric analysis (AA) products suffer from a substantial warm bias of sea-ice skin temperature during clear-sky episodes in Arctic winter when compared to in-situ observations from a recent campaign and infrared satellite observations. They claim that this warm bias is maintained by excessive conductive heat flux through the ice arising from a missing representation of an insulating layer of snow on top of the ice. They support their claim by a simplified calculation of the heat flux under the assumption of constant meteorological conditions, and by analyzing a pair of Arctic regional analyses with and without representation of a snow layer.

General assessment:

I judge the manuscript to be of high interest to specialists in the field, for the following two reasons: (1) Although the producers of atmospheric reanalyses are well aware of the drawbacks of the simplified representation of sea ice and snow layer on top, this has been poorly documented - not least due to a lack of high-quality in-situ observations. Hence, there are many users of AA who treat them as a substitute for observations, without questioning the degree of realism that can be expected depending on the physical parameter under consideration. It is useful to have the limits of AA clearly documented. (2) In recent years, it has become clear that the thickness of the snow layer on top of sea ice is critical for making progress in remote sensing, modeling, assimilating and forecasting of sea-ice related parameters. The current manuscript provides proof that there is a direct detrimental impact on AA if snow on sea ice is not accounted for.

The authors' characterization of the temperature bias is well-supported by the observational data they present, however there is a potential caveat that needs to be discussed (see major comment below). Their conclusion that snow on sea ice is responsible for the temperature bias rests on two independent lines of arguments: results from simplified heat budget calculations and from a pair of regional atmospheric reanalysis with and without a snow layer. The results support the author's conclusion well.

The paper is generally well-written, but it looks like the Methods section has been moved to the end without making the main text self-contained. This creates some disruptions when reading and needs to be fixed.

I would recommend that his manuscript is suitable for publication in Nature Communications if the comments below can be addressed satisfactorily.

Major comment:

The authors do not explicitly state whether the AA data they use is the grid-cell mean skin temperature or the skin temperature over the ice-covered part of the grid cell. I suspect the authors compare the grid-cell mean skin temperature of the atmospheric reanalysis to the skin temperature measured on the ice by the campaign. This would be problematic as it can potentially lead to false conclusions. As evident from the inset in Figure 1, there is a substantial amount of leads in the region of the N-ICE 2015 campaign. These leads have a surface temperature of -2 Celsius when they expose open water, and only slightly lower temperatures when they become covered by a layer of newly formed thin sea ice. If even a small fraction of the relevant AA grid cell is represented as open water, the grid-cell average skin temperature is significantly higher than the skin temperature of the ice. For instance, if the ice skin temperature is -30 Celsius (as in the cold events shown in Figure 2), a lead area fraction of only 10% raises the grid-cell mean skin temperature by 3 K. This is comparable to the magnitude of the bias that the authors present as their central conclusion. Therefore, the authors need to:

1) state explicitly for each AA whether they compare observations to the grid-cell mean skin

temperature (wrong), or the skin temperature of the ice-covered part of the grid cell (correct).
2) In case it is the former, reproduce Figures 2 and 3 with the ice skin temperature T_i from the AA. This can be computed in good approximation for each AA grid cell from the equation $T_{cell} = (1 - SIC) T_w + SIC * T_i$, where T_{cell} is the grid-cell mean temperature, T_w is -2 Celsius, and SIC is the sea-ice concentration used by the AA.
3) If step (2) results in significantly lower AA ice temperatures, the conclusions need to be adapted accordingly.

Minor comments:

- l. 25: add "in the Arctic." to the end of the sentence
- l. 26-33: This paragraph is too generic, the same could be said for any other region of the globe. I suggest to revise or delete.
- l. 59f: Please explain briefly how the skin temperature observations were taken.
- l. 63: experiment AA-snow needs to be introduced before discussing its results.
- l. 71ff: Please clarify whether you are talking about grid-cell mean skin temperature or the skin temperature of the ice-covered part of the grid cell (see major comment).
- l 82ff: These calculations assume a steady state, which might not be a valid assumption for quickly changing meteorological conditions and a thick ice/snow layer. At least some of the AA compute the conductive heat flux based on a vertical discretization of several layers in the ice. The simulated temperatures in these layers could be used to obtain a better approximation of the actually simulated heat flux for any given day in the AA. I do not believe that this would change the conclusions, but the strong simplification of assuming a steady state should be mentioned and discussed to some extent.
- l 129ff: This is related to my major comment, please be explicit about sea-ice concentration used in each AA.
- Figure 1: Is not referenced in the main text.
- Figure 2: Please change the color map chosen, it is very difficult for viewers who are red-green color blind to tell apart negative of positive values. Using the same color map as in Fig 3c-f should be okay, although I would suggest to use white in the middle to denote no error.

Reviewer #2 (Remarks to the Author):

Getting the surface temperature predictions right for Arctic sea ice in atmospheric models is an important goal when investigating Arctic amplification. This paper is relatively straightforward but really convincing in its analysis, namely that a reasonable treatment of the impact of snow cover on sea ice is needed in atmospheric models to model the skin temperature accurately during clear skies in winter. I recommend acceptance subject to some modest improvements.

1. You have looked at a broader class of models than just global reanalyses, Table 1 lists AA and IFS-HIRES that are forecast models. You might want to think about changing the title to include this diversity of atmospheric models but this is not mandatory.
2. Reference should be made to both Figure 1 and Table 1 in the main text, not left to the Methods section.
3. Comments about the reanalyses. Refer to these as global reanalyses as regional reanalyses are becoming more common. NCEP-2 cannot be described as a contemporary reanalysis although it is still widely used. Similarly reference 7 discusses ERA-40 that has been succeeded by two generations of ECMWF reanalyses, ERA-Interim and now ERA5. Good recent references on Arctic reanalysis performance are:
Lindsay, R., M. Wensnahan, A. Schweiger, and J. Zhang, 2014: Evaluation of seven different atmospheric reanalysis products in the arctic. *J. Clim.*, 27, 2588–2606, doi:10.1175/JCLI-D-13-00014.
Cullather, R., T. M. Hamill, D. Bromwich, X. Wu, and P. Taylor, 2016: Systematic Improvements of Reanalyses in the Arctic (SIRTA). A White Paper., <http://www.iarpccollaborations.org/uploads/cms/documents/sirta-white-paper-final.pdf>.
4. Line 89: Change to "extent".
5. Lines 127-129: I found this sentence a little hard to follow when discussing JRA-55 and NCEP-2

temperature biases. Consider rephrasing to improve understandability.

6. Lines 166-167: Spread in climate models in the net longwave radiation budget during winter. Where does this statement come from? Certainly not from Table 2.

7. Line 191: Change to "performed".

8. Table 1: Change "ice" to "sea ice".

Reviewer #3 (Remarks to the Author):

The manuscript presented a warm bias of the sea-ice surface temperature in atmospheric reanalyses over Arctic winter is mainly caused by a missing representation of the snow layer on top of the sea-ice. This is an important finding, since reanalysis products and their Arctic surface energy budget are frequently used for climate model validation, process understanding, forcing of ocean and sea ice models, and initialization and assimilation of seasonal and decadal prediction systems. The finding is also help communities who want to improve the model and reanalyses. I think it perhaps worth to be published in Nature Communications after some revision.

My specific comments are attached as supplementary material.

Reviewer #3 (Remarks to the Author):

Review report for "On the warm bias of atmospheric reanalyses over Arctic sea-ice in winter" written by Yurii Batrak, Malte Müller

The manuscript presented a warm bias of the sea-ice surface temperature in atmospheric reanalyses over Arctic winter is mainly caused by a missing representation of the snow layer on top of the sea-ice. This is an important finding, since reanalysis products and their Arctic surface energy budget are frequently used for climate model validation, process understanding, forcing of ocean and sea ice models, and initialization and assimilation of seasonal and decadal prediction systems. The finding is also help communities who want to improve the model and reanalyses. I think it perhaps worth to be published in Nature Communications after some revision.

Specific comments

- I think the title of “Abstract”, “Introduction” and “References” should be added before L5, L19 and L268, respectively.
- I would not list references in the abstract part between L6 and L16.
- L19-39, it is better to point out the importance of the surface energy budget as you stated in L5-10 in this part, probably after L33.
- L34-36, “in these conditions” is not clear for me. It is in the CSE conditions or no solar radiation and strong longwave radiation conditions? Reword the sentence.
- L40-50, It is better to point out at the start of this paragraph that you use both regional (N-ICE 2015) and pan-Arctic scale data.
- L42, refer to “(Figure 1)” after “between January and June 2015”.
- L45, reword to “N-ICE. We investigate the ability of ...”
- L46-47, It would be nice to add “AA and AA-snow” and refer to “Table 1”, and add “see Methods”.
- L49, may reword part of the sentence ,“In order to check the validity of our findings, we include...”.

Analysis of a single clear-sky event

- L51, I would like to change the title to “Temperature bias over north of Svalbard” to make it consistent with your later section title of “A pan-Arctic view of the temperature bias”. (L97)
- L56, remove “rates”
- L57, reword the sentence to “We refer in the following to three CSEs, CSE-1, CSE-2, and CSE-3, which are also marked in Figure 2.”.
- L59, add references for “the consistent with the observations.”
- L60, better to replace “reach” with another word to clearly show that the simulated surface temperature is higher than the observed low values.
- L62, I would reword it to “5 to 15 °C higher, in particular in the snow-free AA experiment. The sensitivity...”.

- L65-70, reword this paragraph. For me, the model performances are similar during CSE-2 and CSE-3. Maybe first describe the general situation of all the models performance regarding surface temperature and LW during the two CSEs, and then describe the different performance of the models.
- L65, replace “thus, not resolving” with “indicating not well resolved”.
- L73, change “offset” to “difference”
- L79, in Table 1, the ice thickness for MERRA-2 is n/a^a, but here a 1.5 m sea-ice thickness is used for MERRA-2 reanalysis. Will this influence your result? Maybe you can discuss this with your later results about misrepresentation of snow and the sea-ice thickness?
- L82-88, I will start with the AA-snow experiment since it shows the best result.
- L87, Although JRA-55 and NCEP-2 both have a sea ice thickness of 2 m, JRA-55 has no snow but NCEP-2 has snow. Do you think that sea ice plays a more important role than snow for the two reanalyses?

A pan-Arctic view of the temperature bias

- L102-105. I would add (Figure 3c-f) after “about 5 to 10 °C” in L102, and (Figure 3e) after MERRA-2 in L102, (Figure 3c) after ERA5 in L103, (Figure 3d) after JRA-55 in L104, and (Figure 3f) after “NCEP-2” in L105, to help readers to easily understand the text and the figures.
- L103, change “show” to “showing”
- L104, change “has” to “having”
- L109-111, I will combine the two sentences together, starting with “The patterns of the Tskin bias in the reanalyses shows similar characteristics to the retrieved mean snow depth from the TOPAZ4 ocean and sea-ice reanalysis during the CSE events 2015-2017.” to continue with what you are talking in the last paragraph.
- Accordingly rewritten L111-114.
- Is it possible to add one title for the part L115-132 to emphasize the content
- L120-122, rewrite this sentence. I think you can better explain how the error in snow depth resulted in the warm bias in the ERA5 and MERRA-2 in Figure 3c and 3e combining with Figure 3b and 3a.
- L128, add “agrees” or “is consistent” after the “former”.
- L129-131, rewrite the sentence to make it concise.

Summary and conclusions

- L136-137, replace “for the period that is covered by” with “using”
- L138-139, I will drop “it can be consistently shown that”, and add “and reanalyses” after model. I think model and reanalysis are not the same.
- L140-147, is there conflict? Here you point out that the warm bias results from the too high outgoing longwave radiation, and the overestimated conductive heat flux. While in L93-94, you mentioned that the warm bias is only due to the overestimated conductive heat flux, since too high surface temperatures and an increase in outgoing longwave radiation are used to compensate the overestimated conductive heat flux.
- L152, maybe specify what is “previous arguments”
- L152-155, it is difficult to understand for me, rewrite this sentence, or even remove it
- L156, remove “the consideration”

- L158, replace “missing” with “a missed”
- L174, specify “this bias”, it is the temperature bias, or no snow layer or ...?

Method2

- L178, add “and reanalyses” after “Atmospheric models”
- L189, remove “was defined with”
- L191-192, I think it should “January 10”, “March 20”, “September 1”, “March 20”
- L192, replace “this allows” with “allowing”
- L193-194, rewrite this sentence to make it concise and specify “this experiment”, I think it is AA-snow experiment?
- L195, add “and reanalyses” after “the models”
- L198, replace “contributes” with “modifies”
- L198-199, here you say that “NCEP-2 is the only reanalysis which simulates snow sea-ice”, but when you see Table 1, “AA-snow” also has snow on top of sea-ice, clarify this.
- L201, nice to drop “as well”
- L202, better to write as “simulate sea-ice surface temperature”
- L212, “.” is missed at the end of the sentence
- L219-220, it is difficult me to understand this sentence. Reword it.

Observations

- L225, can change “are focusing” to “focus”, and remove “in” before “between”
- L226, add “analyze” before “the meteorological observations”
- L227, remove “observations are analysed”
- L235, I would specify “those events” as “CSEs” to make it clear.

Surface Energy budget and surface temperature

- L244, Eq.(3), I understand that the solar radiation is essentially zero during the polar night. But I think it is better you list the solar radiation, and say the solar radiation is essentially zero, or you have to point out that Eq. (3) is for the surface energy budget during the polar night

Table 1

- legend, add “and reanalyses” after “Model”
- row 2, for “weather model”, I think it is better to call it “atmospheric model and reanalyses”, since you list not only weather model, but also reanalyses.

Response to the Reviewers:

Dear Reviewers.

Thank you very much for the very helpful and constructive reviews. Please see below for a point by point response on your corrections and suggestions. Note that our replies are marked by **blue colour** and indentation.

Reviewer #1

Review of the manuscript "On the warm bias of atmospheric reanalyses over Arctic sea-ice in winter" by Yuri Batrak and Malte Muller.

The authors report that several widely-used atmospheric analysis (AA) products suffer from a substantial warm bias of sea-ice skin temperature during clear-sky episodes in Arctic winter when compared to in-situ observations from a recent campaign and infrared satellite observations. They claim that this warm bias is maintained by excessive conductive heat flux through the ice arising from a missing representation of an insulating layer of snow on top of the ice. They support their claim by a simplified calculation of the heat flux under the assumption of constant meteorological conditions, and by analyzing a pair of Arctic regional analyses with and without representation of a snow layer.

General assessment:

I judge the manuscript to be of high interest to specialists in the field, for the following two reasons: (1) Although the producers of atmospheric reanalyses are well aware of the drawbacks of the simplified representation of sea ice and snow layer on top, this has been poorly documented - not least due to a lack of high-quality in-situ observations. Hence, there are many users of AA who treat them as a substitute for observations, without questioning the degree of realism that can be expected depending on the physical parameter under consideration. It is useful to have the limits of AA clearly documented. (2) In recent years, it has become clear that the thickness of the snow layer on top of sea ice is critical for making progress in remote sensing, modeling, assimilating and forecasting of sea-ice related parameters. The current manuscript provides proof that there is a direct detrimental impact on AA if snow on sea ice is not accounted for.

The authors' characterization of the temperature bias is well-supported by the observational data they present, however there is a potential caveat that needs to be discussed (see major comment below). Their conclusion that snow on sea ice is responsible for the temperature bias rests on two independent lines of arguments: results from simplified heat budget calculations and from a pair of regional atmospheric reanalysis with and without a snow layer. The results support the author's conclusion well.

The paper is generally well-written, but it looks like the Methods section has been moved to the end without making the main text self-contained. This creates some disruptions when reading and needs to be fixed.

I would recommend that his manuscript is suitable for publication in Nature Communications if the comments below can be addressed satisfactorily.

Major comment:

The authors do not explicitly state whether the AA data they use is the grid-cell mean skin temperature or the skin temperature over the ice-covered part of the grid cell. I suspect the authors compare the grid-cell mean skin temperature of the atmospheric reanalysis to the skin temperature measured on the ice by the campaign. This would be problematic as it can potentially lead to false conclusions. As evident from the inset in Figure 1, there is a substantial amount of leads in the region of the N-ICE 2015 campaign. These leads have a surface temperature of -2 Celsius when they expose open water, and only slightly lower temperatures when they become covered by a layer of newly formed thin sea ice. If even a small fraction of the relevant AA grid cell is represented as open water, the grid-cell average skin temperature is significantly higher than the skin temperature of the ice. For instance, if the ice skin temperature is -30 Celsius (as in the cold events shown in Figure 2), a lead area fraction of only 10% raises the grid-cell mean skin temperature by 3 K. This is comparable to the magnitude of the bias that the authors present as their central conclusion.

Thank you very much for this important comment. We indeed compared the grid-cell mean skin temperature derived from the outgoing and incoming long-wave radiation with the one measured over sea-ice during N-ICE 2015. During the time period we analyse the three clear sky events, the sea-ice concentration has been larger than 90%. Specifically during the first event (22-Jan-2015 to 2-Feb-2015), which we analyse in detail in this study, the concentration has been 100%. Please see figure below for the sea-ice concentration along the N-ICE drift.

Therefore, the authors need to:

1) state explicitly for each AA whether they compare observations to the grid-cell mean skin temperature (wrong), or the skin temperature of the ice-covered part of the grid cell (correct).

We explicitly describe in the text, that:

1. For the N-ICE 2015 comparison (Figure 2) for all models, besides NCEP-2 and JRA-55, the sea-ice surface skin temperature products are used. In those systems this is a prognostic variable from the upper most layer of the sea-ice model. For NCEP-2 and JRA-55 we are using the outgoing and incoming longwave radiation, and the formula is given in the Method section. For NCEP-2 and JRA-55 there is no effect from the ocean on the grid-cell mean surface temperature, since these systems have a binary sea-ice representation with thresholds larger than 55%.
2. For the pan-Arctic comparison (Figure 3) with the satellite measurements we still believe that the grid-cell mean temperature is the correct model output to compare with. The comparison is performed in the areas where sea-ice is classified as closed sea-ice, i.e. sea-ice concentration larger than 80%. The instrument measures the surface temperature without distinguishing between sea-ice and ocean and thus should be consistent with the grid-cell mean skin temperature from the models.

2) In case it is the former, reproduce Figures 2 and 3 with the ice skin temperature T_i from the AA. This can be computed in good approximation for each AA grid cell from the equation $T_{cell} = (1 - SIC) T_w + SIC * T_i$, where T_{cell} is the grid-cell mean temperature, T_w is -2 Celsius, and SIC is the sea-ice concentration used by the AA.

Thank you for providing this useful approximation. However, since the sea-ice surface temperature is available as a direct output from the sea-ice models, we utilize this source of information. As described above, for JRA55 and NCEP-2 it is not necessary to do this

distinction. Figure 2 has been updated accordingly. For Figure 3, as discussed above, we still use the grid-cell mean temperature due to its consistency with the satellite measurements.

3) If step (2) results in significantly lower AA ice temperatures, the conclusions need to be adapted accordingly.

Step 2, had a small impact on Figure 2 and Table 2. However, it had no impact on the overall results and conclusion. Results in Figure 3 are unchanged.

Minor comments:

- l. 25: add "in the Arctic." to the end of the sentence

Added

- l. 26-33: This paragraph is too generic, the same could be said for any other region of the globe. I suggest to revise or delete.

We rewrote this paragraph and put it in context to the main topics of the paper.

- l. 59f: Please explain briefly how the skin temperature observations were taken.

We included this information in the Method section and Equation 3.

- l. 63: experiment AA-snow needs to be introduced before discussing its results.

We updated the manuscript to introduce AA and AA-snow experiments before discussing the results.

- l. 71ff: Please clarify whether you are talking about grid-cell mean skin temperature or the skin temperature of the ice-covered part of the grid cell (see major comment).

We describe in the text which surface temperature product has been used from the model systems, as well as, for the N-ICE 2015 observations.

- l. 82ff: These calculations assume a steady state, which might not be a valid assumption for quickly changing meteorological conditions and a thick ice/snow layer. At least some of the AA compute the conductive heat flux based on a vertical discretization of several layers in the ice. The simulated temperatures in these layers could be used to obtain a better approximation of the actually simulated heat flux for any given day in the AA. I do not believe that this would change the conclusions, but the strong simplification of assuming a steady state should be mentioned and discussed to some extent.

We fully agree that steady state assumption is not generally valid, especially for multilayer schemes. However, in our study we focus on clear sky events during the polar night and in these conditions variations of the atmospheric state are not so drastic. We added a note about steady state assumption to the Methods section and mentioned in the main text that conductive heat flux was calculated assuming steady state.

- l. 129ff: This is related to my major comment, please be explicit about sea-ice concentration used in each AA.

We describe in the text which surface temperature product has been used for the comparison with the infrared satellite measurements.

- Figure 1: Is not referenced in the main text.

We added the missing reference to the main text.

- Figure 2: Please change the color map chosen, it is very difficult for viewers who are red-green color blind to tell apart negative of positive values. Using the same color map as in Fig 3c-f should be okay, although I would suggest to use white in the middle to denote no error.

We updated the colormap of Figure 2 to avoid green and red on the same figure. To denote areas with small errors a light-yellow color is used to distinguish them from the areas with missing data.

Reviewer #2:

Getting the surface temperature predictions right for Arctic sea ice in atmospheric models is an important goal when investigating Arctic amplification. This paper is relatively straightforward but really convincing in its analysis, namely that a reasonable treatment of the impact of snow cover on sea ice is needed in atmospheric models to model the skin temperature accurately during clear skies in winter. I recommend acceptance subject to some modest improvements.

1. You have looked at a broader class of models than just global reanalyses, Table 1 lists AA and IFS-HIRES that are forecast models. You might want to think about changing the title to include this diversity of atmospheric models but this is not mandatory.

Thank you for this relevant comment. We decided to not change the title, but to give more emphasis on this fact in the abstract by writing “global atmospheric reanalyses and weather forecasts” instead of “atmospheric analyses”.

2. Reference should be made to both Figure 1 and Table 1 in the main text, not left to the Methods section.

We added references to Figure 1 and Table 1 to the main text.

3. Comments about the reanalyses. Refer to these as global reanalyses as regional reanalyses are becoming more common. NCEP-2 cannot be described as a contemporary reanalysis although it is still widely used. Similarly reference 7 discusses ERA-40 that has been succeeded by two generations of ECMWF reanalyses, ERA-Interim and now ERA5. Good recent references on Arctic reanalysis performance are:

Lindsay, R., M. Wensnahan, A. Schweiger, and J. Zhang, 2014: Evaluation of seven different atmospheric reanalysis products in the arctic. *J. Clim.*, 27, 2588–2606, doi:10.1175/JCLI-D-13-00014.

Cullather, R., T. M. Hamill, D. Bromwich, X. Wu, and P. Taylor, 2016: Systematic Improvements of Reanalyses in the Arctic (SIRTA). A White Pap.,.

<http://www.iarpccollaborations.org/uploads/cms/documents/sirta-white-paper-final.pdf>.

Thank you for the valuable comment and references. We updated the text to explicitly mention that we focus on global reanalysis products, where applicable.

4. Line 89: Change to “extent”.

Changed

5. Lines 127-129: I found this sentence a little hard to follow when discussing JRA-55 and NCEP-2 temperature biases. Consider rephrasing to improve understandability.

We reworded that sentence to more understandable and added references to the Figure which shows bias maps of JRA-55 and NCEP-2.

6. Lines 166-167: Spread in climate models in the net longwave radiation budget during winter. Where does this statement come from? Certainly not from Table 2.

Indeed, the reference to Sorteberg et al. 2007 was missing and has been added.

7. Line 191: Change to “performed”.

Changed

8. Table 1: Change “ice” to “sea ice”.

Changed

Reviewer #3:

The manuscript presented a warm bias of the sea-ice surface temperature in atmospheric reanalyses over Arctic winter is mainly caused by a missing representation of the snow layer on top of the sea-ice. This is an important finding, since reanalysis products and their Arctic surface energy budget are frequently used for climate model validation, process understanding, forcing of ocean and sea ice models, and initialization and assimilation of seasonal and decadal prediction systems. The finding is also help communities who want to improve the model and reanalyses. I think it perhaps worth to be published in Nature Communications after some revision.

Specific comments

- I think the title of “Abstract”, “Introduction” and “References” should be added before L5, L19 and L268, respectively.

We included this information.

- I would not list references in the abstract part between L6 and L16.

We removed the reference in the abstract section.

- L19-39, it is better to point out the importance of the surface energy budget as you stated in L5-10 in this part, probably after L33.

We included a paragraph in the Introduction to point out the importance of the surface energy budget.

- L34-36, “in these conditions” is not clear for me. It is in the CSE conditions or no solar radiation and strong longwave radiation conditions? Reword the sentence.

We reworded the sentence to make it more clear.

- L40-50, It is better to point out at the start of this paragraph that you use both regional (N-ICE 2015) and pan-Arctic scale data.

We reorganized this paragraph accordingly.

- L42, refer to “(Figure 1)” after “between January and June 2015”.

We added the missing reference to the Figure 1

- L45, reword to “N-ICE. We investigate the ability of ...”

We re-worded accordingly.

- L46-47, It would be nice to add “AA and AA-snow” and refer to “Table 1”, and add “see Methods”.

We added a brief introduction of AA and AA-snow experiments and a reference to the Table 1.

- L49, may reword part of the sentence ,“In order to check the validity of our findings, we Include...”.

The paragraph and the sentence has been reorganized following to an earlier comment.

Analysis of a single clear-sky event

- L51, I would like to change the title to “Temperature bias over north of Svalbard” to make it consistent with your later section title of “A pan-Arctic view of the temperature bias”. (L97)

The title has been changed to “Analysis of a clear-sky event from N-ICE 2015”

- L56, remove “rates”

Removed

- L57, reword the sentence to “We refer in the following to three CSEs, CSE-1, CSE-2, and CSE-3, which are also marked in Figure 2.”.

Thank you for the comment. We re-worded accordingly.

- L59, add references for “the consistent with the observations.”

We added a reference to the Figure 2.

- L60, better to replace “reach” with another word to clearly show that the simulated surface temperature is higher than the observed low values.

We reworded the sentence to make it clear that simulated surface temperature is higher than observed during N-ICE 2015 drift campaign.

- L62, I would reword it to “5 to 15 °C higher, in particular in the snow-free AA experiment. The sensitivity...”.

We reworded to: “... most of the models simulate temperatures of 5 to 15 C too warm, with the strongest deviations found in the snow-free AA experiment. The sensitivity experiment AA-snow, with a prognostic sea-ice thickness and snow layer model, ... “

- L65-70, reword this paragraph. For me, the model performances are similar during CSE-2 and CSE-3. Maybe first describe the general situation of all the models performance regarding surface temperature and LW during the two CSEs, and then describe the different performance of the models.

We agree that this paragraph was not well structured and modified the next according to the comment.

- L65, replace “thus, not resolving” with “indicating not well resolved”.

We modified the text accordingly.

- L73, change “offset” to “difference”

We replaced “offset” by “discrepancy” to make the sentence more clear.

- L79, in Table 1, the ice thickness for MERRA-2 is n/aa, but here a 1.5 m sea-ice thickness is used for MERRA-2 reanalysis. Will this influence your result? Maybe you can discuss this with your later results about misrepresentation of snow and the sea-ice thickness?

We recalculated the conductive heat flux for MERRA-2 with assumed ice thicknesses of 1 and 2 meter. The flux are 61 and 30 W/m², respectively. The result is in line with our arguments and has no implications for our results and conclusions. We added this information to the text.

- L82-88, I will start with the AA-snow experiment since it shows the best result.

We modified the text accordingly

- L87, Although JRA-55 and NCEP-2 both have a sea ice thickness of 2 m, JRA-55 has no snow but NCEP-2 has snow. Do you think that sea ice plays a more important role than snow for the two reanalyses?

Indeed there is a snow layer over sea ice in NCEP-2, but this snow layer parameterization is very basic and has no effect on the thermal conductivity of the sea ice and does not contribute to surface albedo. Thus, snow on sea ice in NCEP-2 could be safely neglected. Some details about snow cover in NCEP-2 are provided in Table 1 and Methods section. To not confuse the readers we added a note mentioning that snow cover in NCEP-2 could be neglected.

A pan-Arctic view of the temperature bias

- L102-105. I would add (Figure 3c-f) after “about 5 to 10 °C” in L102, and (Figure 3e) after MERRA-2 in L102, (Figure 3c) after ERA5 in L103, (Figure 3d) after JRA-55 in L104, and (Figure 3f) after “NCEP-2” in L105, to help readers to easily understand the text and the figures.

We modified the text accordingly

- L103, change “show” to “showing”

Corrected.

- L104, change “has” to “having”

Corrected.

- L109-111, I will combine the two sentences together, starting with “The patterns of the Tskin bias in the reanalyses shows similar characteristics to the retrieved mean snow depth from the TOPAZ4 ocean and sea-ice reanalysis during the CSE events 2015-2017.” to continue with what you are talking in the last paragraph.

We modified the sentences accordingly.

- Accordingly rewritten L111-114.

We are not sure what the reviewer is referring to with this comment. This text part remains unchanged

- Is it possible to add one title for the part L115-132 to emphasize the content

We considered to subdivide this section of the manuscript. However, we feel that the last part of the section fits very well to the title “A pan-Arctic view of the temperature bias”, since it explains the pan-Arctic temperature characteristics of the model products and put it in context with a snow thickness reanalysis product and the assumptions of sea-ice thicknesses and sea-ice concentration in the models.

- L120-122, rewrite this sentence. I think you can better explain how the error in snow depth resulted in the warm bias in the ERA5 and MERRA-2 in Figure 3c and 3e combining with Figure 3b and 3a.

We reworded that sentence and added references to the Figure 3 to make it more readable and understandable.

- L128, add “agrees” or “is consistent” after the “former”.

We reworded that sentence to make it more readable.

- L129-131, rewrite the sentence to make it concise.

We rewrote this sentence to make it more clear and understandable.

Summary and conclusions

- L136-137, replace “for the period that is covered by” with “using”

Corrected according to suggestion

- L138-139, I will drop “it can be consistently shown that”, and add “and reanalyses” after model. I think model and reanalysis are not the same.

The sentence has been reworded.

- L140-147, is there conflict? Here you point out that the warm bias results from the too high outgoing longwave radiation, and the overestimated conductive heat flux. While in L93-94, you mentioned that the warm bias is only due to the overestimated conductive heat flux, since too high surface temperatures and an increase in outgoing longwave radiation are used to compensate the overestimated conductive heat flux.

There is no conflict. We write that the too high longwave radiation is a consequence of the warm bias, and not vice versa. It has also been shown in detail in Table 2 and the analysis of CSE-1 in a previous section, that this too high long wave outgoing radiation balances the missing snow layer. Furthermore, the longwave outgoing radiation is directly connected to the surface temperature by the Stefan-Boltzman law.

- L152, maybe specify what is “previous arguments”

We specified that we mean that the snow layer is connected to the very low temperatures and the temperature bias.

- L152-155, it is difficult to understand for me, rewrite this sentence, or even remove it

We reworded the sentence to make it more clear.

- L156, remove “the consideration”

corrected

- L158, replace “missing” with “a missed”

We think ‘missing’ is more appropriate in this sentence.

- L174, specify “this bias”, it is the temperature bias, or no snow layer or ...?

We included “temperature”

Method

- L178, add “and reanalyses” after “Atmospheric models”

We changed the name of this subsection to “Atmospheric models and reanalysis products”

- L189, remove “was defined with”

Corrected

- L191-192, I think it should “January 10”, “March 20”, “September 1”, “March 20”

We corrected the dates to be consistent within the manuscript and to follow the British English variant of spelling.

- L192, replace “this allows” with “allowing”

We reworded that sentence

- L193-194, rewrite this sentence to make it concise and specify “this experiment”, I think it is AA-snow experiment?
We reworded the confusing sentence and added more details about initialization of AA and AA-snow experiments.
- L195, add “and reanalyses” after “the models”
This information has been included.
- L198, replace “contributes” with “modifies”
We replaced it by “impacts”.
- L198-199, here you say that “NCEP-2 is the only reanalysis which simulates snow sea-ice”, but when you see Table 1, “AA-snow” also has snow on top of sea-ice, clarify this.
We agree that this is misleading, and thus we removed “ the only reanalysis which is”.
- L201, nice to drop “as well”
Corrected.
- L202, better to write as “simulate sea-ice surface temperature”
Corrected.
- L212, “.” is missed at the end of the sentence
We added the missing full stop mark.
- L219-220, it is difficult me to understand this sentence. Reword it.
Reworded to improve readability

Observations

- L225, can change “are focusing” to “focus”, and remove “in” before “between”
Corrected.
- L226, add “analyze” before “the meteorological observations”
Corrected.
- L227, remove “observations are analysed”
Corrected.
- L235, I would specify “those events” as “CSEs” to make it clear.
Modified.

Surface Energy budget and surface temperature

- L244, Eq.(3), I understand that the solar radiation is essentially zero during the polar night. But I think it is better you list the solar radiation, and say the solar radiation is essentially zero, or you have to point out that Eq. (3) is for the surface energy budget during the polar night
We extended the equation to include the short-wave radiation flux and pointed that solar radiation does not contribute to the surface energy budget during the polar night.

Table 1

- legend, add “and reanalyses” after “Model”
We extended the title of the Table 1 to mention that it provides information on both numerical weather prediction systems and global reanalysis systems.

- row 2, for “weather model”, I think it is better to call it “atmospheric model and reanalyses”, since you list not only weather model, but also reanalyses.

We think that “weather model” is more appropriate there because that row provides information about the weather models which were applied to run numerical experiments and produce corresponding global reanalysis products.

Reviewers' comments:

Reviewer #1 (Remarks to the Author):

I am mostly satisfied with how the authors responded to my concerns. In particular, I am now more convinced that their findings are not invalidated by incorrect usage of reanalysis output.

However, in their response and the revised manuscript, some contradictions remain that should be clarified:

1) In ll. 84ff of the revised manuscript, the authors state that "T_{skin} is directly obtained from the respective sea-ice or snow models and thus is only representing the sea-ice covered part of each grid cell". This is in contradiction to their response to my major comment, where they state "We indeed compared the grid-cell mean skin temperature derived from the outgoing and incoming long-wave radiation..." These statements are in contradiction, please clarify.

2) To my knowledge, the skin temperature over the sea-ice covered part of the grid cell is not available from at least one of the reanalyses considered. I had a look at the ERA5 data documentation on <https://confluence.ecmwf.int/display/CKB/ERA5+data+documentation>. There is a skin temperature available (short name skt, parameter ID 235), but according to the IFS documentation that is the grid-cell mean. There is another parameter called "Ice temperature layer 1" (short name istl1, parameter ID 35) - this is indeed valid for the sea-ice covered part of the grid cell, but it is a layer temperature (upper 7cm) and not a skin temperature. Can the authors please clarify which one they used? In case they use parameter skt, my initial comment holds, and it cannot be directly compared with in-situ measurements over ice in cases where sea-ice concentration in the grid cell is below 100%. In case the study uses parameter istl1, this is not a skin but a layer temperature, which will always be warmer than the skin during cold spells. Hence, both cases could in principle give a warm bias because of diagnostic inconsistencies, even if the modelled skin temperature over the ice was in perfect agreement with observations. As stated above, I do not believe that the author's findings are too much affected by inconsistencies in how they use the model fields. But, given the strong statements they make about deficiencies in the reanalysis, any ambiguities should be resolved, and remaining uncertainties (e.g. if approximations were made to diagnose the sea-ice skin temperature from available model output) should be clearly stated.

Reviewer #3 (Remarks to the Author):

I only have one comment to the authors: Regarding my comments "L59, add References for the 'the consistent With the observations'". The Authors replied that they added a Reference to the Figure 2. But I cannot find where they added the Reference. Maybe I overlooked.

Otherwise, I am satisfied With the replies of the Authors to my comments.

Response to the Reviewers:

Dear Reviewers.

Thank you very much for the response to the first revision of our paper. Please see below for a point by point response on your corrections and suggestions. Note that our replies are marked by **blue colour** and indentation.

Reviewer #1

I am mostly satisfied with how the authors responded to my concerns. In particular, I am now more convinced that their findings are not invalidated by incorrect usage of reanalysis output.

However, in their response and the revised manuscript, some contradictions remain that should be clarified:

1) In ll. 84ff of the revised manuscript, the authors state that "Tskin is directly obtained from the respective sea-ice or snow models and thus is only representing the sea-ice covered part of each grid cell". This is in contradiction to their response to my major comment, where they state "We indeed compared the grid-cell mean skin temperature derived from the outgoing and incoming long-wave radiation..." These statements are in contradiction, please clarify.

Sorry for the confusion. In the original submitted version we calculated the Tskin "from the outgoing and incoming long-wave radiation" and we changed this in the 1st revision to the Tskin taken "from the respective sea-ice or snow models". The sentence in the response was misleading and did not refer to the revised version of the manuscript.

2) To my knowledge, the skin temperature over the sea-ice covered part of the grid cell is not available from at least one of the reanalyses considered. I had a look at the ERA5 data documentation on <https://confluence.ecmwf.int/display/CKB/ERA5+data+documentation>. There is a skin temperature available (short name skt, parameter ID 235), but according to the IFS documentation that is the grid-cell mean. There is another parameter called "Ice temperature layer 1" (short name istl1, parameter ID 35) - this is indeed valid for the sea-ice covered part of the grid cell, but it is a layer temperature (upper 7cm) and not a skin temperature. Can the authors please clarify which one they used? In case they use parameter skt, my initial comment holds, and it cannot be directly compared with in-situ measurements over ice in cases where sea-ice concentration in the grid cell is below 100%. In case the study uses parameter istl1, this is not a skin but a layer temperature, which will always be warmer than the skin during cold spells. Hence, both cases could in principle give a warm bias because of diagnostic inconsistencies, even if the modelled skin temperature over the ice was in perfect agreement with observations. As stated above, I do not believe that the author's findings are too much affected by inconsistencies in how they use the model fields. But, given the strong statements

they make about deficiencies in the reanalysis, any ambiguities should be resolved, and remaining uncertainties (e.g. if approximations were made to diagnose the sea-ice skin temperature from available model output) should be clearly stated.

We fully agree that the temperature of the upper most sea ice layer often differs from the real skin temperature. Hence, we modified the computation of T_{skin} and follow your advice from the 1st revision, to partition the grid cell. We partition the long wave outgoing radiation according to the sea-ice concentration at the grid cell. The T_{skin} over the sea-ice part of the grid cell is then calculated from the sea-ice related outgoing longwave radiation part. This is described in detail in the updated Methods section (lines 270 -283). The Figure below shows the impact of those changes in case of the ERA5 reanalysis. The green line corresponds to the T_{skin} of the original submitted manuscript version, i.e. from the in- and out-going longwave radiation. The magenta line is the temperature of the upper ice-layer, from the 1st revision. The black line is the T_{skin} which corresponds to the sea-ice part only. It shows that there is a systematic effect of these changes, however, compared to the bias to the observations (red line) the changes are still small. Figure 2 has been updated accordingly. Since sea-ice concentration was close to 100% during the first CSE, the results in Table 2 are not impacted.

Reviewer #3

I only have one comments to the authors: Regarding my comments "L59, add References for the "the consistent With the observations". The Authors replied that they added a Reference to the Figure 2. But I cannot find where they added the Reference. Maybe I overlooked.

Indeed, we referred to the Figure 2 in that sentence as it was indicated in our response. Although, the reference was located at the end of the sentence and could be easily overlooked. We moved the reference to a more appropriate position in the middle of the sentence to make it more clear and visible.

Otherwise, I am satisfied with the replies of the Authors to my comments.

REVIEWERS' COMMENTS:

Reviewer #1 (Remarks to the Author):

I am satisfied with how the authors addressed my previous comments and recommend publication.

Response to the Reviewers:

Dear Reviewer.

Thank you very much for the positive response

Reviewer #1 (Remarks to the Author):

I am satisfied with how the authors addressed my previous comments and recommend publication.